# Quality of care index and gender disparity ratio for stroke and its subtypes from the Global Burden of Disease Study 2021

Aqian Yu[1,2], Jingjing An[1,2]*

1 West China School of Nursing, Sichuan University, Chengdu, China, 2 Operating Room, Department of Anesthesiology, West China Hospital, Sichuan University/West China School of Nursing, Sichuan University, Chengdu, Sichuan, China

* jingjinganwch@126.com

## Abstract

### Objectives

This study aims to explore the quality of care index (QCI) for stroke, intracerebral hemorrhage (ICH), subarachnoid hemorrhage (SAH), and ischemic stroke (IS) across nations, genders, age groups, and periods from the Global Burden of Diseases Study (GBD) 2021.

### Methods

Prevalence, incidence, and years lived with disability (YLDs) rates were used to estimate the burden. Estimated annual percentage change (EAPC) was utilized to evaluate trends. QCI was calculated through principal component analysis. Gender disparity ratio (GDR), a ratio of female QCI and male QCI, was used to evaluate the disparity. Spearman correlation analysis was performed to investigate the association between QCI, GDR, and SDI.

### Results

The age-standardized prevalence, incidence, and YLDs rates for stroke, ICH, SAH, and IS decreased from 1990 to 2021 (EAPC<0). The QCI showed an increased trend for stroke (51.6 vs 65.7) worldwide. The GDR for stroke, ICH, and IS improved (near one), whereas the GDR of IS experienced a slight increase. The values of QCI increased with the rising of SDI for stroke, ICH, SAH, and IS. However, the values of GDR decreased with the increasing of SDI for stroke, ICH, and IS. Considerable disparities between countries, genders, and age groups concerning the QCI and GDR were observed.

**Data availability statement:** The data can be accessed openly through the GBD 2021 (https://vizhub.healthdata.org/gbd-results/).

**Funding:** The author(s) received no specific funding for this work.

**Competing interests:** The authors have declared that no competing interests exist.

## Conclusion

The QCI and GDR improved from 1990 to 2021 among most countries and regions. Both QCI and GDR were associated with SDI. Significant disparities between countries, genders, and age groups for QCI and GDR should be addressed.

---

## Introduction

Stroke stands as the third most fatal disease (7.3 million [95% UI 6.6–7.8] deaths; 10.7% [9.8–11.3] of all deaths) and ranks fourth in contributing to the burden in total disability-adjusted life years (DALYs) (160.5 million [147.8–171.6] DALYs; 5.6% [5.0–6.1] of all DALYs) worldwide [1]. Although substantial progress in prevention, treatment, and prognosis has been made, the number of strokes and related deaths surged markedly from 1990 (50.4 million) to 2021 (93.8 million). It is estimated that 84.1% (77.8 to 88.8) of stroke DALYs were attributable to the risk factors. A cross-sectional study in China found that the estimated prevalence, incidence, and mortality rate of stroke among adults aged 40 years or older in 2020 were 2.6%, 505.2 per 100 000 person-years, and 343.4 per 100 000 person-years, respectively [2]. These suggest that stroke management has not yielded the expected results. The Lancet Neurology Commission projections reveal a stark upward trajectory in the global toll of stroke: mortality from this devastating condition is anticipated to surge by 50%—climbing from 6.6 million lives lost (with a 95% uncertainty interval (UI) of 6.0 million to 7.1 million) in 2020 to 9.7 million (8.0 million to 11.6 million) by 2050 [3]. Globally, the financial toll of stroke—encompassing both direct expenses (such as acute medical treatment and long-term rehabilitation services) and indirect costs (stemming from lost productivity as patients and caregivers are sidelined from work)—is estimated to surpass an astonishing US$891 billion each year [4]. As he four pillars of the stroke quadrangle (surveillance, prevention, acute care, and rehabilitation), acute care and rehabilitation are essentials in stroke management [3]. This finding casts a clear, urgent light on the imperative of sustaining unwavering attention to elevating the quality of care for those living with stroke.

Some metrics, such as prevalence, incidence, and mortality, can reflect stroke burden and changes separately. However a more comprehensive understanding of stroke quality assessments requires additional indicators, such as the quality of care, to assess timely interventions and effective treatment [5]. The definition for quality of care by the World Health Organization was the extent to which health services for individuals and populations enhance the probability of achieving desired health outcomes [6]. It reflects the delivery of meticulously tailored, medical interventions designed to align with individual patient needs while maximizing therapeutic efficacy and enhancing long-term health prospects. High level of quality of care is crucial in managing stroke. In addition, the inequality of quality of care among countries was intensified [7]. Great disparities in stroke burden by GBD region, country or territory, sex, and SDI were observed. For example, some countries in the European Region showed substantial differences in morbidity, prevalence, mortality, and DALYs rate

[8]. Quality of care, comparing with prevalence, incidence, and mortality, may be a more comprehensive indicator to represent the difference in stroke care. The Global Burden of Disease 2021 provides the latest and most valuable data concerning stroke burden and changes from 1990 to 2021 [9]. Crude or even age-standardized rates are inadequate for capturing the true quality of care in GBD 2021. Previous studies have explored the schizophrenia [10], multiple sclerosis [11], and low back pain [12]. However, comparing the quality of care for stroke across regions and countries was absent. Therefore, we derive the quality of care index (QCI) to quantify healthcare quality and reveal critical inequities in service provision across territory and age demographics by applying principal component analysis to GBD health metrics [13]. In this study, we tried to use the quality of care index (QCI) to measure and compare the stroke burden and trends between countries, genders, and age groups, and explore the potential causes using the data from GBD 2021, thereby guiding therapeutic strategies making and policy frameworks.

## Methods

All data used in this study are publicly available on the GBD website. Informed consent was not needed as the study did not involve patient-specific data. Ethical reviewed from ethical review board was exempted. All maps were reprinted from Resource and Environmental Science Data Platform [https://www.resdc.cn], under a CC BY license, with permission from the Institute of Geographic Sciences and Natural Resources Research, Chinese Academy of Sciences, original copyright © 2014–2026.

### Stroke data sources

In GBD 2021, cross-sectional data for stroke and its subtypes, including intracerebral hemorrhage (ICH), subarachnoid hemorrhage (SAH), and ischemic stroke (IS) were obtained from the Global Health Data Exchange, which includes the estimates of 371 diseases and injuries among 21 regions and 204 countries and territories, from 1990 to 2021 [14]. Countries were categorized into five SDI-based tiers: low, low-middle, middle, high-middle, and high [15]. The estimated prevalence and incidence rates were evaluated through the DisMod-MR 2.1 method. YLDs were calculated by multiplying the prevalence of varying severity levels by their corresponding disability weight. More details were displayed in **Supplementary Methods**.

### Statistical analysis

A descriptive analysis displayed the stroke burden on global, regional, and national scales. We used the estimated annual percentage change (EAPC) to explore the changing trends from 1990 to 2021 [16]. Generally, statistically significant changes typically reflect EAPC estimates and their 95% confidence intervals (CIs) above or below zero. However, the 95% CIs in EAPC encompassed zero, indicating no significant changes over time.

The QCI was calculated with the combination of the available primary indices (prevalence, incidence, mortality, years of life lost (YLLs), YLDs, and DALYs) and four introduced meaningful secondary indices. (1) The mortality-to-incidence ratio (MIR), (2) DALY-to-prevalence ratio, (3) Prevalence-to-incidence ratio, and (4) YLLs -to-YLDs ratio [13]. The four formulas are described below:

$$MIR = \frac{\text{mortality}}{\text{incidence}}$$

$$DPR = \frac{DALYs}{Prevalence}$$

$$PIR = \frac{Prevalence}{Incidence}$$

$$YLR = \frac{YLLs}{YLDs}$$

The YLD metric evaluates the equilibrium between life years curtailed by untimely death and the diminishment in life quality caused by impaired health. An elevated YLD ratio reveals a disease's tendency to claim lives prematurely, while a diminished ratio reflects its primary impact as a chronic burden that erodes vitality without markedly shortening lifespan [17]. A higher DPR extracts a heavy toll on those afflicted, while a subdued ratio reflects ailments a more prevalent diseases with a milder health impact [18]. The MIR quantifys disease lethality by calculating the proportion of fatal outcomes among diagnosed cases. This indicator functions as a proxy for healthcare system performance, where a reduced MIR value demonstrates superior clinical management and therapeutic efficacy [19]. The PIR represents an analytical measure that evaluates the temporal dynamics of disease by quantifying the association between condition duration and case emergence frequency. Elevated PIR values are pathognomonic for chronic disease processes, indicating prolonged disease states with sustained population prevalence [18].

Then, a principal component analysis (PCA), which is a statistical procedure to reduce data dimensionality by deriving linear combinations from various datasets and preserving most variation [20], was performed to transfer these four indices into a summary measure. The original study utilized principal component analysis selected the first principal component (PC1) as a surrogate for the QCI due to its inherent capacity to explain the greatest proportion of variance in the dataset [21]. We developed an enhanced QCI by combining variance-weighted PC1 and PC2. This refined metric demonstrates superior sensitivity in discriminating care quality levels compared to traditional single-component approaches, with higher values indicating better healthcare performance [22]. The QCI ranges from 0 to 100, and the higher score represents the better the quality of health care (Table 1).

$$PC1 = \omega1 * YLR + \omega2 * DPR + \omega3 * MIR + \omega4 * PIR$$

$$PC2 = \omega5 * YLR + \omega6 * DPR + \omega7 * MIR + \omega8 * PIR$$

The weighting parameters $\omega_1$ through $\omega_8$ correspond to the eigenvector elements associated with the largest eigenvalues in the PCA decomposition, mathematically representing each variable's loading on the principal component subspace. PC1 constitutes the optimal one-dimensional projection maximizing explained variance ($\lambda_1$), with PC2 providing the subsequent orthogonal projection direction ($\lambda_2$) that optimally captures residual variance under the constraint of component orthogonality.

$$PCAscore(x) = \left(\frac{Var(PC1)}{(Var(PC1) + Var(PC2))}\right) * PC1 + \left(\frac{Var(PC2)}{(Var(PC1) + Var(PC2))}\right) * PC2$$

In this formulation, $Var(PC_1)$ denotes the eigenvalue-associated variance captured by the first principal component. $Var(PC_2)$ corresponds to the residual variance accounted for by the second orthogonal component.

**Table 1. The QCI and GBR.**

| Metric | significance | explanation |
|---|---|---|
| QCI | the quality of health care | ranges from 0 to 100, and the higher score represents the better the quality of health care |
| GDR | the gender disparity of QCI | values above 1 suggest relatively higher stroke QCI in women, and values below 1 indicate relatively higher stroke QCI in men |

$$QCI(x) = \frac{PCAscore(x) - minPCAscore(x)}{maxPCAscore(x) - minPCAscore(x)} * 100\%$$

In this formula, $QCI(x)$ represents the Quality Care Index for a given observation x, with higher scores indicating superior care quality. $PCA_{sore}(x)$ represents the principal component score for observation x derived from our composite metric.

Further validation of the QCI results for stroke in the present study was undertaken using the Healthcare Access and Quality (HAQ) Index, developed by the GBD study collaborators to assess quality of care (QoC) and access to healthcare across a range of medical conditions [23]. A linear mixed-effects regression model was employed, with the stroke QCI serving as the dependent variable and mean inpatient and outpatient healthcare utilization [24], risk factor exposure, and stroke-related mortality as independent variables.

We computed the gender disparity ratio (GDR) to compare gender inequality. GDR is defined as the ratio of female QCI to male QCI. A ratio approaching one signified a higher degree of equality. However, values above 1 suggest relatively higher stroke QCI in women, and values below 1 indicate relatively higher stroke QCI in men (Table 1).

Spearman correlation analysis is conducted to explore the association between QCI, GDR, and SDI. This method calculates the ρ indices and P values. $P < 0.05$ is an indicator of statistical significance.

## Results

### Epidemiology of stroke

Globally, stroke and its subtypes, including ICH, SAH, and IS, accounted for more than 15.2 million, 2.7 million, 1.2 million, and 11.3 million YLDs in 2021, respectively (S1 Table in S2 File). By contrast, the age-standardized YLDs rates for stroke, ICH, SAH, and IS were significantly decreased for stroke and its subtypes (EAPC<0), respectively.

Based on the SDI classification, the age-standardized YLDs rates for stroke, ICH, SAH, and IS showed a declining trend (EAPC<0), except for an increase for IS in countries with a high-middle SDI (EAPC = 0.38). The age-standardized prevalence and incidence rates decreased for stroke, ICH, SAH, and IS (EAPC<0), whereas an upward trend of prevalence and incidence was detected for IS in countries with a middle SDI.

At regional levels, the age-standardized YLDs rates for stroke, ICH, SAH, and IS decreased from 1990 to 2021, except for an increase for stroke in East Asia, for ICH in high-income North America, for SAH in high-income Asia Pacific, for IS in East Asia. Only the age-standardized incidence rates of IS in East Asia increased (S2 and S3 Tables in S2 File).

In different ages, assessment of stroke, ICH, SAH, and IS burden in 2021 followed a pattern of increasing trend globally with age onward (Fig 1).

### QCI

The global age-standardized QCI for stroke, ICH, SAH, and IS increased (S4 Table in S2 File). QCI in all 5 SDI regions experienced an increase for both sexes, females, and males (Fig 2). As for the regions in 2021, the highest age-standardized QCI for stroke, ICH, SAH, and IS were in high-income North America (stroke, QCI = 99.8), high-income Asia Pacific (ICH, QCI = 95.2), high-income Asia Pacific (SAH, QCI = 91.0), high-income North America (IS, QCI = 99.6) for both sexes and the same locations were observed for females and males (S1–S4 Figs in S1 File). The QCI for stroke and IS in Southern Sub-Saharan Africa, ICH in Central and Southern Sub-Saharan Africa, and SAH in Central Latin America, Central Asia, and Southern Sub-Saharan Africa experienced a decline for both sexes, females, and males from 1990 to 2021 (S5 Fig in S1 File).

At the national level, United States of America for stroke (QCI = 100.0) and IS (QCI = 100.0), Singapore (QCI = 100.0) for ICH, and Kuwait for SAH (QCI = 100.0) had the highest age-standardized QCI in both sexes in 2021 (S5 Table in S2 File, Fig 3), and the same nations were observed for females and males, except for the highest age-standardized QCI of SAH for both sexes and females in Kuwait and the highest age-standardized QCI of SAH for males in Singapore (S6 and S7 Figs

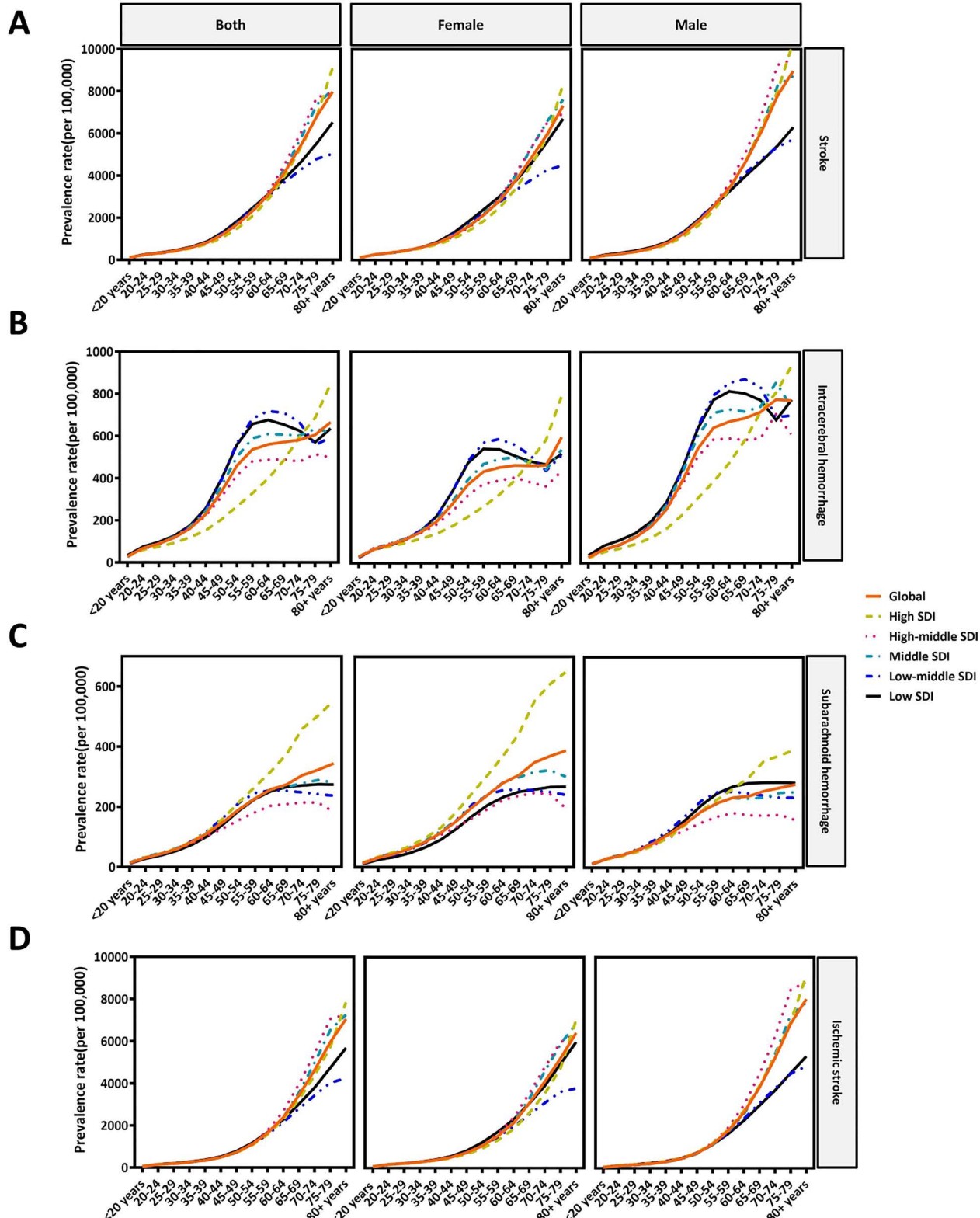

**Fig 1. The age-standardized prevalence rate of stroke(A), ICH (B), SAH (C), and IS (D) in global and 5 SDI regions in different age groups in both sexes, female and male in 2021.**

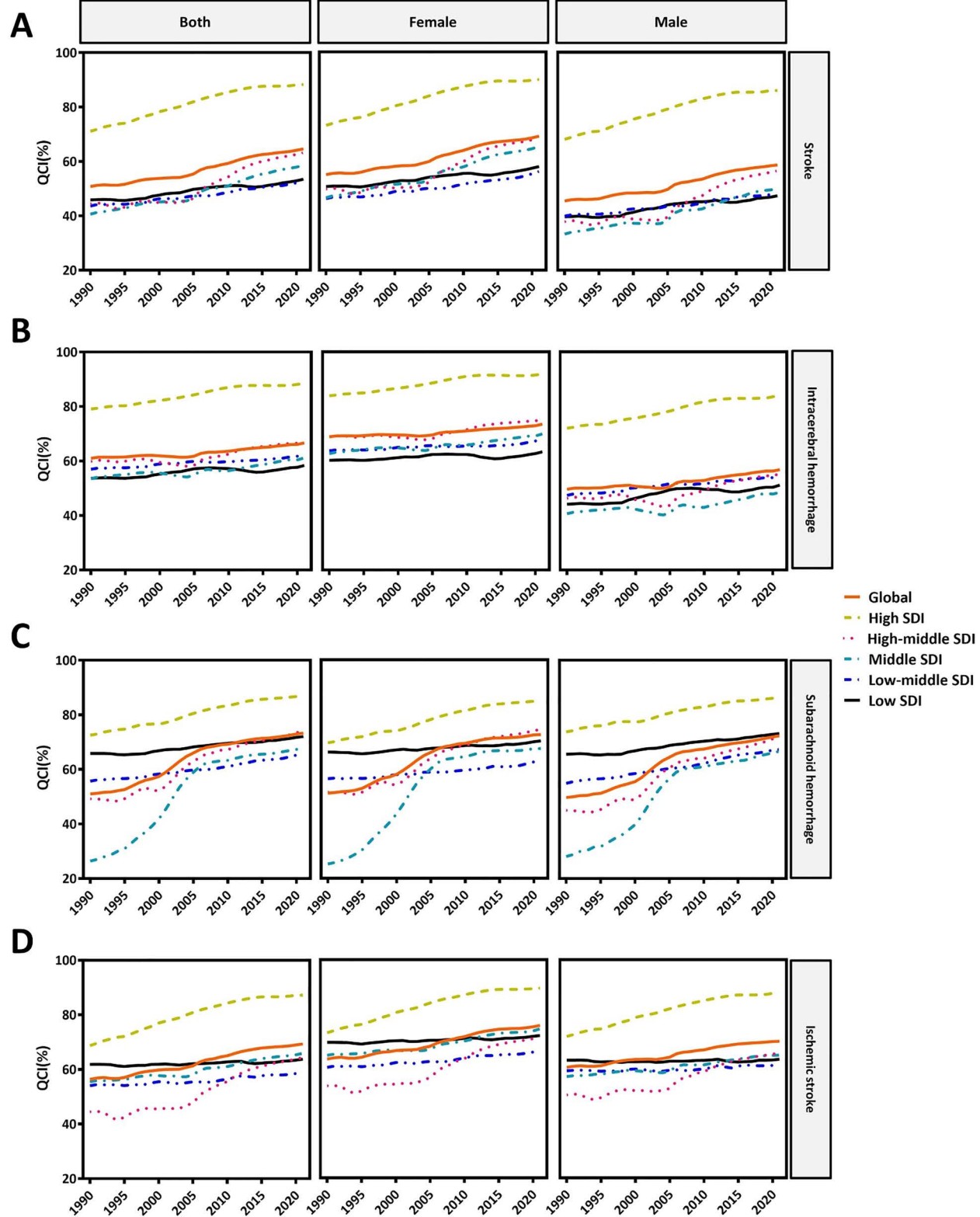

**Fig 2. Time trend of age-standardized QCI for stroke (A), ICH (B), SAH (C), and IS (D) from 1990 to 2021 in global and 5 SDI regions in both sexes, female, and male.**

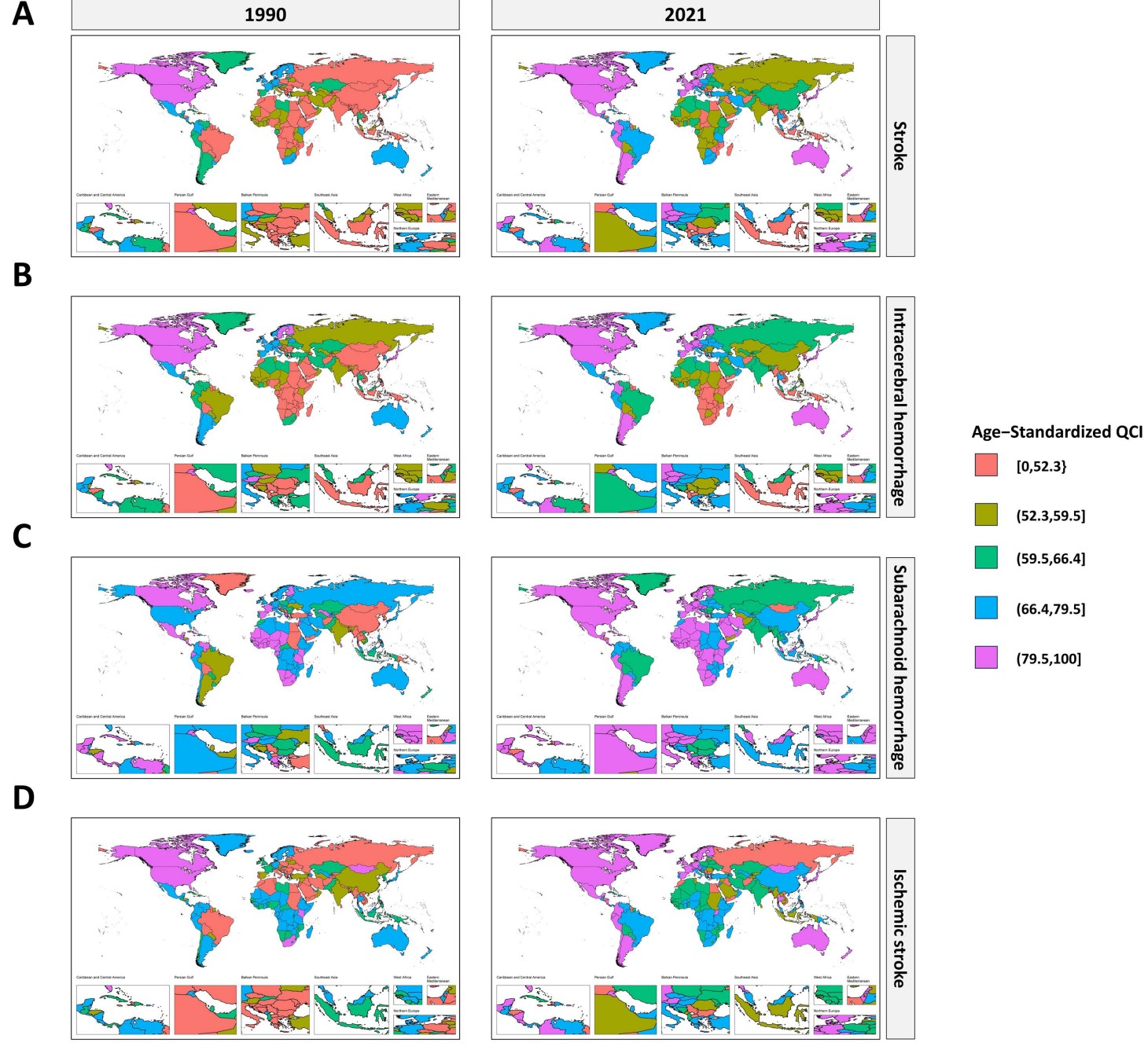

**Fig 3. Geographical distribution of age-standardized QCI for stroke, ICH, SAH, and IS in 1990 and 2021 for both sexes.**

in S1 File). The number of growing QCI countries was higher among men than women for ICH in 1990 and 2021, but not for stroke and IS (S6 Table in S2 File, S8–S12 Figs in S1 File). In both sexes, the QCI for stroke and IS in Honduras, ICH in Zimbabwe, SAH in Uzbekistan experienced the highest decline between 1990 and 2021, whereas QCI for stroke and ICH in Egypt, for SAH in China, and IS in Estonia had the most significant improvement in both sexes (S13–S17 Figs in S1 File).

At global and SDI regional levels, we found that the QCI values increased from 1990 to 2021 in both sexes, females, and males, and countries with a high SDI had a higher QCI compared with other regions (S7–S9 Tables in S2 File). In stroke, the QCI firstly decreased and then touched the bottom at 65–69 years, and then increased in global and 5 SDI regions (S18 Fig in S1 File). A similar trend was observed for IS. More obvious fluctuations were observed in men in all age groups compared with women in both 1990 and 2021.

There was an increased trend of QCI with the SDI rose among 204 nations (S19 Fig in S1 File). The QCI of stroke and ICH increased with SDI. The QCI of SAH and IS increased with SDI after SDI exceeding 0.75 (Fig 4).

## Percent changes of QCI

In the percent change analysis from 1990 to 2021, QCI in stroke, SAH, and IS had a more obvious increase in men than in women at global and 5 SDI levels, whereas the opposite situation was observed in ICH (S10 Table in S2 File). Southern Sub-Saharan Africa for stroke (QCI = 29.2), South Asia for ICH (QCI = 46.6), high-income Asia Pacific for SAH (QCI = 100.0), and East Asia for IS (QCI = 88.3) had the highest age-standardized QCI for both sexes in 2021. Overall, the increasing QCIs in the percent changes were higher for men than for women in 21 regions (S11 Table in S2 File).

The increase of QCI for stroke, SAH, and IS in most countries was greater for men than women, except for ICH (S20 and S21 Figs in S1 File, S12 Table in S2 File). Stroke in Indonesia, ICH in Tajikistan, SAH in Uzbekistan, and IS in Vanuatu had the highest age-standardized QCI. Overall, the number of countries for QCIs in stroke and IS was higher for men than women in percent change analysis. By contrast, the number of countries for percent changes of the higher QCIs in SAH was higher for women than men (202 vs 2).

## GDR

The global age-standardized GDR for stroke, ICH, and SAH decreased from 1990 to 2021, whereas the GDR for IS increased from 1.05 to 1.08. Overall, the GDR for stroke and ICH showed a decreased trend (near one) in all SDI regions from 1990 to 2021, whereas the GDR of IS revealed an upward trend (away one) (S13 Table in S2 File, S22 Fig in S1 File). A ratio of away one for GDR was in countries with low SDI (from 1.01 to 0.96), low-middle SDI (from 1.03 to 0.94) for SAH, however, GDR in high-middle SDI region (from 1.14 to 1.03), middle SDI region (from 0.90 to 1.01), and high SDI region (from 0.94 to 0.99) showed a ratio of near one, reflecting the equality between men and women.

In 21 GBD regions, countries in East Asia had the highest GDRs for stroke and ICH both in 1990 and 2021 (stroke, 1.85 and 1.39; ICH, 1.95 and 1.56), and North Africa and Middle East countries in 199 (GDR = 1.16) for SAH and East Asia countries for SAH in 2021 had the highest GDRs (1.08 in 2021). The highest GDRs for IS were in Western Sub-Saharan Africa in 1990 and 2021. Overall, the number of regions for GDR with a ratio of near one (between 0.95 and 1.05) increased from 1990 to 2021 for stroke, ICH, SAH, and IS (S14 Table in S2 File and S23 Fig in S1 File), suggesting the equality between men and women.

Overall, there has also been an increase in the number of countries that show a ratio of near one for GDR for stroke, ICH, and SAH, but not for IS. The countries with the highest QCI for ICH (Singapore) and SAH (Kuwait) in 2021, showed some of the closest GDR scores to one with ratios of 1.00 and 1.02, respectively, but not for stroke and IS. In 2021, stroke in Lesotho, ICH in Zimbabwe, SAH in Mozambique, and IS in Guinea-Bissau were the highest GDR (S24 Fig in S1 File).

Stroke in Haiti, ICH in Zimbabwe, SAH in Greenland, and IS in Egypt had the highest increase in GDR from 1990 to 2021 (S25 Fig in S1 File), whereas stroke in Guyana, ICH in Afghanistan, SAH in Qatar, and IS in Russian Federation had the highest decline. The number of countries for GDR that did not show a ratio of near one for GDR decreased from 1990 to 2021 for stroke, ICH, and SAH, except for IS. GDR for stroke and ICH in most countries worldwide exceeded one in 1990 and 2021, and the value of GDR for SAH and IS in most countries is approximately 1 (S26 Fig in S1 File), indicating

   

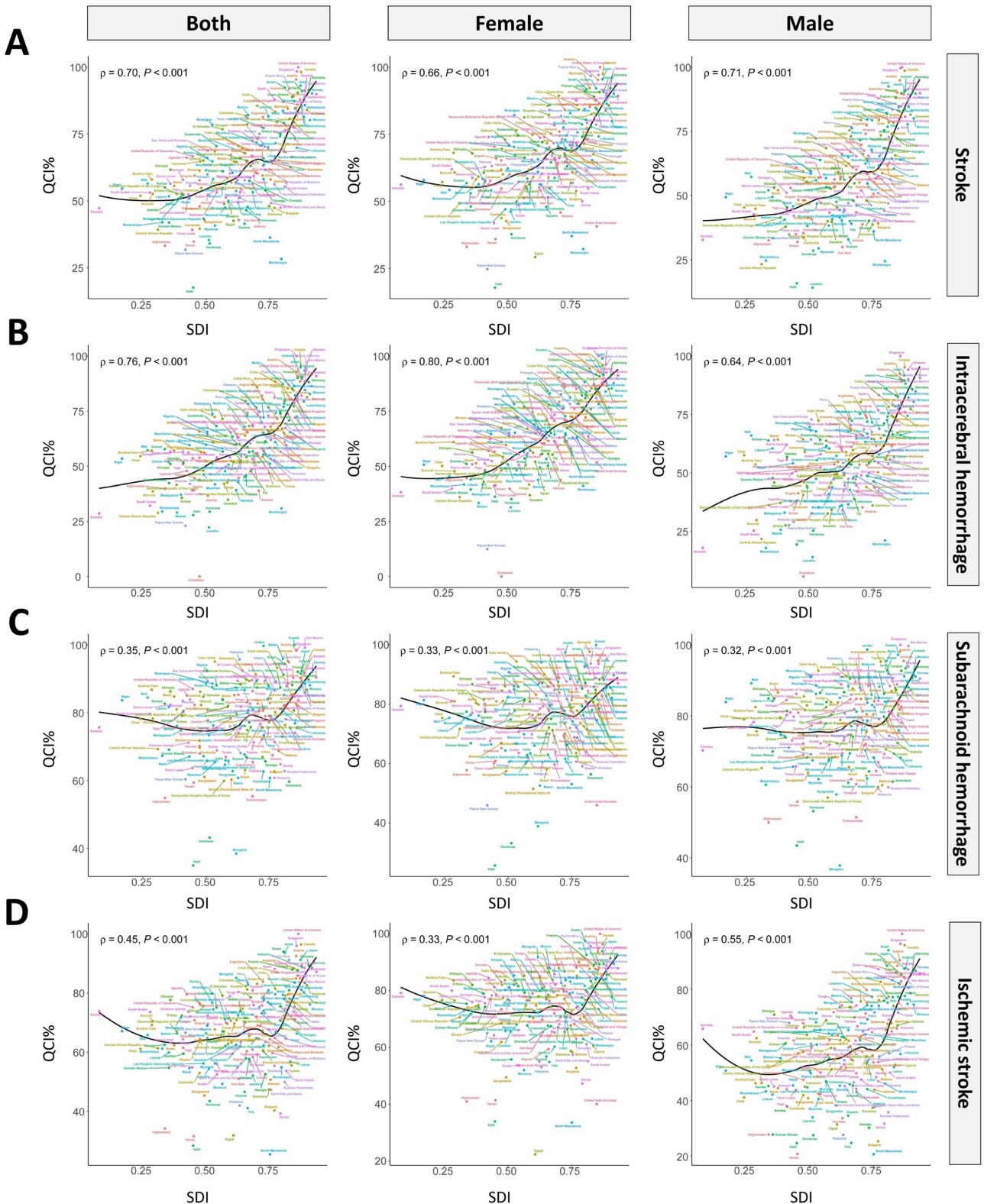

**Fig 4. Age-standardized QCI of stroke (A), ICH (B), SAH (C), and IS (D) by SDI in 2021 for 204 countries and territories against their SDIs for both sexes, females, and males.**

the equality between men and women. More details for the GDR of individual countries for stroke, ICH, SAH, and IS from 1990 to 2021 are shown in S15–16 Tables in S2 File and S27 Fig in S1 File.

With the growth of SDI, GDR was gradually near one, except for SAH (S28 Fig in S1 File). The values of QCI increased with the rising of SDI in 204 nations, except for the GDR of SAH (Fig 5).

The GDR in over 65 years old for stroke, and IS experienced a significant increase in 2021 in global and 5 SDI regions compared to GDR in 1990 (S29 Fig in S1 File and S17 Table in S2 File). The number of regions and countries for GDR near one in different ages at GBD regional and national levels increased from 1990 to 2021 (S18–S21 Tables in S2 File).

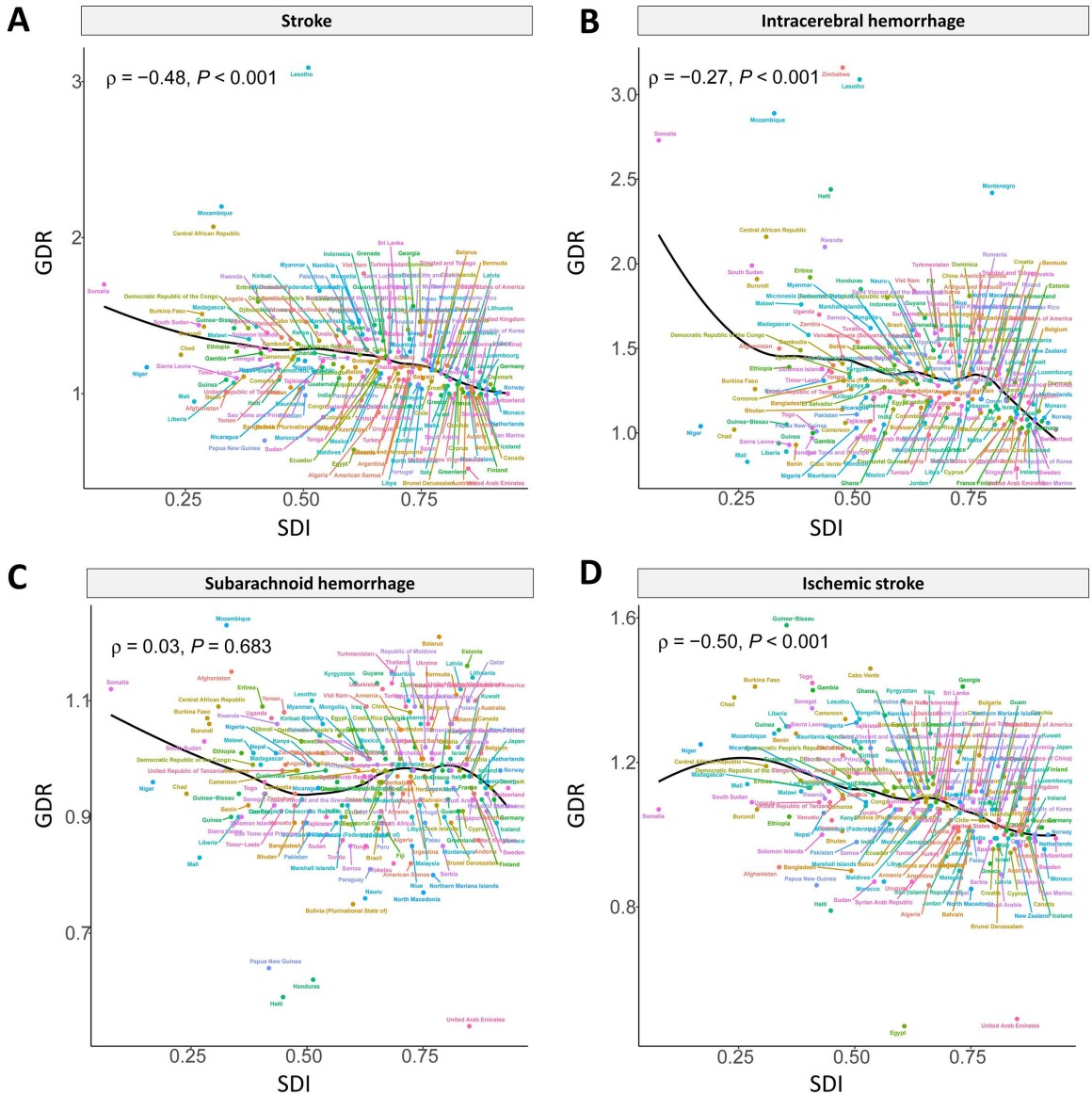

**Fig 5. Age-standardized GDR of stroke (A), ICH (B), SAH (C), and IS (D) by SDI in 2021 for 204 countries and territories against their SDIs.**

### Stroke burden in China

The ASPR, ASIR, ASMR, and DALYs rate of stroke in China is 1301.4, 204.8, 138.0, and 2648.0 per 100,000 persons, respectively. The ASPR, ASIR, ASMR, and DALYs rate of ICH in China is 222.1, 61.2, 68.8, and 1351.6 per 100,000 persons, respectively. The ASPR, ASIR, ASMR, and DALYs rate of IS in China is 1018.8, 135.8, 64.5, and 1181.0 per 100,000 persons, respectively. The ASPR, ASIR, ASMR, and DALYs rate of SAH in China is 68.9, 7.8, 4.7, and 115.5 per 100,000 persons, respectively (S22 Table in S2 File).

We found that the stroke and IS ASPR and IS ASIR had increasing trends (EAPC>0) from 1990 to 2021 (S23 Table in S2 File).

## Discussion

### Global burden

The increasing burden in China was one of the main reasons leading to the rising trend of stroke in East Asia. Except for common risk factors, the rapid advancement of aging has increased the stroke burden in China [25]. Therefore, the Chinese government has made substantial endeavors to enhance stroke prevention and treatment [26]. Aging also impacted the global stroke burden. A study found that aging is an important factor leading to the increasing stroke burden caused by non-optimal temperatures [27]. The increasing number of deaths was closely related to aging in the past three decades [28]. The increasing burden of ICH in the United States of America (USA) was one of the main reasons resulting in the rising trend of ICH in high-income North America. In line with our findings, a nationwide America study found that the prevalence of ICH increased by 11% which was mainly driven by young and middle-aged people [29]. Hypertension and high anticoagulant use rates in hospitalized patients were associated with this increase. The increase in SAH burden in Japan was a main factor leading to the upward trend in high-income Asia Pacific from 1990 to 2021. It has been reported that SAH incidence has increased by 59.1% in the last three decades in Japan [30]. Compared with the stroke burden in China among Chinese adults aged 40 years and older reported by Wen-Jun Tu et al, the results in GBD were higher in stroke ASPR, ASIR, ASMR, and DALYs rates in age-standardized rates including all age groups [2]. Our study showed that the stroke and IS ASPR and IS ASIR in China had increasing trends from 1990 to 2021.

### QCI

Many risk factors can be modified in the prevention of stroke [31]. The five leading risk factors for stroke were high systolic blood pressure, high body-mass index, high fasting plasma glucose, ambient particulate matter pollution, and smoking [32]. Advancements in treatment strategies across pre-reperfusion, reperfusion, and post-reperfusion phases, along with enhanced access to acute stroke interventions, have markedly improved stroke outcomes [33]. In the management of ICH, the findings from the third Intensive Care Bundle with Blood Pressure Reduction in Acute Cerebral Hemorrhage Trial (INTERACT3) showed the effectiveness of the care bundle for prognosis improvement, including early lowering of elevated blood pressure, strict glucose control, antipyrexia treatment, and rapid reversal of warfarin-related anticoagulation [34].

The decreasing burden of stroke observed in countries with a high SDI, one of the important reasons is the better standards of health care. Low- and middle-income countries suffer shortages in those areas. Similar results have been reported in the rural-urban disparity [35]. The QCI value in women was higher than in men, which might be related to more heavier stroke burden in men [32]. Compared to women, men tend to exhibit poorer management of stroke risk factors, and men's willingness to seek medical attention is also lower [36]. It should be noted that inconsistent results from many retrospective studies concerning SAH functional outcomes were obtained [37]. In addition, some demographic factors, such as age, race, and ethnicity in SAH outcomes, might also impact the difference in QCI between men and women [38].

On the background of aging worldwide, women were usually older at the time of their first stroke occurrence and may have more comorbidities, such as hypertension and atrial fibrillation [39], which also partially explained the poorer

outcomes compared with men [40]. In addition, a study in Denmark found that women underwent less reperfusion therapy than men due to a longer time delay from symptom onset to stroke unit arrival [41]. Another nationwide study in Germany including 1.11 million cases found similar results [42]. Substantive differences in non-focal and focal stroke symptoms between men and women in stroke might impact the effect of treatment and result in misdiagnosis [38].

## GDR

Studies have uncovered that women face more severe outcomes than men after a stroke, encompassing higher mortality rates, reduced quality of life, greater incidence of post-stroke depression, and more significant activity limitations [43]. However, most research focused on a specific age group (mostly in the elderly) or a specific area, and the poor outcome in women could be observed. The insufficient attention in women may intensify in most regions and countries. On the one hand, differences in age, stroke characteristics, and cerebrovascular risk factors at baseline may explain the disparities between sexes. In addition, unfavorable outcomes may be offset by a lower prevalence in women. Women are generally more proactive than men in seeking medical care, whether for physical or psychological concerns, they tend to do that not only before hospitalization but also as part of their post-hospitalization follow-up [36]. This heightened engagement reflects a greater tendency toward preventive health behaviors and a sustained commitment to recovery in higher healthcare utilization, which can be crucial in managing both acute and chronic conditions and may lead to improved long-term outcomes in female stroke survivors.

In countries with a low SDI, the stroke burden is disproportionally increasing [32]. In less developed regions, stroke mortality rises with improvements in socioeconomic status among the group aged 45–54 years, whereas in more developed areas, it declines as socioeconomic status advances [44]. By contrast, higher socioeconomic status was associated with higher stroke rates in people over 75 years [45]. With the increase of SDI, the GDR value gradually was located near one.

## Strengths and limitations

Our results can help develop stroke provision strategies (either local or national) and provide evidence to guide health policy-making procedures in the future. As such, the importance of pursuing preventive and intervention stroke in all regions of the world is reiterated. Prioritising the quality of care for stroke in deprived areas while enhancing access to primary healthcare in these areas is also a key takeaway from our results, emphasizing the need for equitable resource distribution across sex and countries to promote the improvement of care quality for women and backward areas. Future strategies must harmonize specialized stroke interventions with systemic quality improvements across stroke subtypes, advancing toward equitable global care standards. More research with a specific focus on the quality of care for stroke is highly recommended.

We first performed an analysis to evaluate the QCI and gender difference for stroke, ICH, SAH, and IS at the global, regional, and national levels. However, several limitations should be mentioned. First, the GBD data was not stratified by ethnicity. It has been suggested that race can be one of the determining factors influencing the quality of care in stroke [46]. Second, data in GBD was the estimates of the results of complex modeling strategies. However, they were widely accepted data and offered the most comprehensive estimates of epidemiological features of diseases. Third, attribution of QCI and GDR is indirect. Third, due to lack of the HAQ (Healthcare Access and Quality Index) publicly in 2021, we are unable to validate our results. Fourth, it is important to emphasize that the QCI primarily reflects health outcomes and does not directly measure structural or process-related dimensions of healthcare delivery, such as adherence to stroke treatment guidelines, hospital readmission rates, or patient-reported outcomes. However, the QCI is constructed from four well-established epidemiological ratios, all of which have been widely used in previous studies as indirect indicators of healthcare effectiveness, survival, disease management, and disability burden [47–50]. Moreover, the index may be influenced by factors beyond the healthcare system itself, including underlying population health status, disease ecology,

and the completeness and accuracy of disease surveillance systems. Due to the limited availability of comprehensive, longitudinal data covering multiple decades and diverse geographic settings, empirical validation of the QCI against direct and independent indicators of healthcare quality remains constrained. Therefore, future research should prioritize the validation and refinement of the QCI as more granular and reliable data on healthcare delivery become available, which will further enhance its robustness and utility as a proxy measure of healthcare quality. Fifth, although the study correlates QCI/GDR with the SDI, it does not adjust for granular factors within SDI tiers (e.g., healthcare funding, workforce density, or health insurance coverage). However, SDI is a well-established composite indicator developed within GBD framework [51–53]. The use of SDI in this study was intended to examine broad sociodemographic gradients in quality of care rather than to isolate the independent effects of specific healthcare system components. Importantly, the primary aim of this study was to characterize global, regional, and national patterns and disparities in stroke care quality, rather than to establish causal relationships between healthcare system inputs and quality outcomes. Given the global scope of the analysis and the inclusion of 204 countries and territories over multiple decades, consistent and comparable data on granular healthcare system variables are not uniformly available across all locations and time periods. Future studies incorporating more detailed healthcare system indicators will be essential to further disentangle the relative contributions of specific healthcare resources and policies to variations in care quality.

## Conclusion

The QCI and GDR improved from 1990 to 2021 among most countries and regions. Both QCI and GDR were associated with SDI. However, significant disparities between countries, genders, and age groups for QCI and GDR should be addressed. Effective measures, such as the improvement stroke recognition, investment in regional stroke units, and enhancement of guideline adherence monitoring in low-SDI countries among women should be taken to improve the significant disparities.

## Supporting information

**S1 File. Supplementary Figures.**
(PDF)

**S2 File. Supplementary tables.**
(PDF)

**S3 File. Gather-checklist.**
(PDF)

## Acknowledgments

We acknowledge the contributions from GBD 2021.

## Author contributions

**Conceptualization:** Aqian Yu.

**Data curation:** Aqian Yu.

**Investigation:** Aqian Yu, Jingjing An.

**Methodology:** Aqian Yu, Jingjing An.

**Project administration:** Aqian Yu, Jingjing An.

**Visualization:** Jingjing An.

**Writing – original draft:** Aqian Yu.

**Writing – review & editing:** Jingjing An.

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
