## [Decision Letter · Decision Letter 0]

14 Jul 2025

Dear Dr. an,

Thank you for submitting your manuscript to PLOS ONE. After careful consideration, we feel that it has merit but does not fully meet PLOS ONE’s publication criteria as it currently stands. Therefore, we invite you to submit a revised version of the manuscript that addresses the points raised during the review process.

**ACADEMIC EDITOR:**

 2. GBD showed that The annual number of strokes and deaths due to stroke increased substantially from 1990 to 2021. Data on the burden of stroke in your county should be increased so that readers can understand the need for stroke-related studies. Additional citations could be added: Estimated Burden of Stroke in China in 2020. JAMA Netw Open. 2023;6(3): e231455. 3. . This reviewer would expect to see some points regarding how to translate these observations to help address this public health concern. 

We look forward to receiving your revised manuscript.

Kind regards,

Wen-Jun Tu

Academic Editor

PLOS ONE

Journal Requirements:

https://journals.plos.org/plosone/s/file?id=ba62/PLOSOne_formatting_sample_title_authors_affiliations.pdf....

Reviewers' comments:

Reviewer's Responses to Questions

**Comments to the Author**

1. Is the manuscript technically sound, and do the data support the conclusions?

Reviewer #1: Partly

2. Has the statistical analysis been performed appropriately and rigorously?

Reviewer #1: Yes

3. Have the authors made all data underlying the findings in their manuscript fully available?

Reviewer #1: Yes

4. Is the manuscript presented in an intelligible fashion and written in standard English?

Reviewer #1: No

Reviewer #1: General Comment

This manuscript presents an important analysis of the QCI and GDR for stroke and its major subtypes, leveraging data from the GBD Study 2021. The topic aligns well with current global health priorities, particularly the pursuit of equitable and effective care delivery. However, the manuscript would benefit from stronger conceptual framing, more precise reporting of methodological procedures, and clarification of some key terms and interpretations. Specific comments are presented.

Specific comments

Comment 1. The authors refer to the QCI as a composite index derived through PCA based on four secondary metrics. However, a clearer theoretical rationale for how this index captures “quality of care” is needed. What dimensions does it intend to reflect? How has it been validated across conditions or settings?

Comment 2: The GDR is defined as the ratio of female-to-male QCI. A ratio close to 1 is assumed to reflect equity, but this may oversimplify interpretability. What does a GDR >1 or <1 indicate in real-world terms? Consider illustrating implications with country-level examples where GDR diverges notably from parity.

Comment 3: The analysis presents decreasing YLDs alongside increasing QCI and improved GDR. Please clarify how much of the improvement in QCI reflects actual system performance versus changes in stroke incidence/severity. The limitations of indirect attribution need emphasis.

Comment 4: While the results are rich in numerical detail, the manuscript would benefit from a concise interpretive table outlining how to read QCI and GDR values, including thresholds for high/low quality or parity gaps. This would enhance accessibility for both technical and policy audiences.

Comment 5: Sentences such as “The situation intensifies in most regions and countries with the increase in age” (line 267) or “There was a positive association between QCI and SDI” (Abstract) are vague or grammatically unclear. Please clarify.

Comment 6: While age-specific results are reported, the manuscript stops short of interpreting why older age groups exhibit different QCI or GDR patterns. Are care gaps among the elderly driven by biological vulnerability, health system neglect, or cultural inequities? A deeper exploration could enhance interpretive value.

Comment 7: Phrases like “measures should be taken to improve disparities” (line 297) are too general. Consider offering more grounded recommendations, such as improving stroke recognition in women, investing in regional stroke units, or enhancing guideline adherence monitoring in low-SDI countries.

Comment 8: Given the emphasis on disparities, integrating a brief discussion of how health systems might inadvertently reinforce or mitigate inequities, especially in gender and aging populations, could enrich the policy relevance of the manuscript.

.

Reviewer #1: No

---

## [Author Response · Author response to Decision Letter 1]

20 Oct 2025

Response to the Reviewers’ Comments

Dear Reviewers,

We really appreciate all of your constructive notes. All of them have helped us improve the quality of this manuscript significantly. Please find our revised manuscript in the attachments where we have addressed your notes. In the following sections, we provide point-by-point responses to your concerns.

Best regards,

Jingjing An,

MD MHA

ACADEMIC EDITOR:

1. 1. What has previously been published on this topic and what does this work add to the existing literature?

Response: Many thanks for your precious advice.

We have added a content your mentioned in line 60-90 in manuscript-revised.

2. GBD showed that The annual number of strokes and deaths due to stroke increased substantially from 1990 to 2021. Data on the burden of stroke in your county should be increased so that readers can understand the need for stroke-related studies. Additional citations could be added: Estimated Burden of Stroke in China in 2020. JAMA Netw Open. 2023;6(3): e231455.

Response: Many thanks for your valuable advice.

We have added a content your mentioned in line 44-47 in manuscript-revised.

3. This reviewer would expect to see some points regarding how to translate these observations to help address this public health concern.

Response: Many thanks for your constructive advice.

We have added a content your mentioned in line 360-370 in manuscript-revised.

Reviewers' comments:

Reviewer's Responses to Questions

Comments to the Author

1. Is the manuscript technically sound, and do the data support the conclusions?

Reviewer #1: Partly

Response: Many thanks for your professional advice.

We have revised the manuscript carefully.

2. Has the statistical analysis been performed appropriately and rigorously?

Reviewer #1: Yes

Response: Thanks for your feedback! And many thanks for sharing your time and energy in reviewing our work.

3. Have the authors made all data underlying the findings in their manuscript fully available?

Reviewer #1: Yes

Response: Thanks for your feedback! And many thanks for sharing your time and energy in reviewing our work.

4. Is the manuscript presented in an intelligible fashion and written in standard English?

Reviewer #1: No

Response: Many thanks for your professional advice.

We have polished our manuscript.

5. Review Comments to the Author

Reviewer #1: General Comment

This manuscript presents an important analysis of the QCI and GDR for stroke and its major subtypes, leveraging data from the GBD Study 2021. The topic aligns well with current global health priorities, particularly the pursuit of equitable and effective care delivery. However, the manuscript would benefit from stronger conceptual framing, more precise reporting of methodological procedures, and clarification of some key terms and interpretations. Specific comments are presented.

Response: Thanks for your feedback! And many thanks for sharing your time and energy in reviewing our work.

Specific comments

Comment 1. The authors refer to the QCI as a composite index derived through PCA based on four secondary metrics. However, a clearer theoretical rationale for how this index captures “quality of care” is needed. What dimensions does it intend to reflect? How has it been validated across conditions or settings?

Response: Many thanks for your professional advice.

We have added a content your mentioned in line 113-171 in manuscript-revised.

The Healthcare Access and Quality Index (HAQ) in 2019 has been provided in https://ghdx.healthdata.org/record/ihme-data/gbd-2019-healthcare-access-and-quality-1990-2019. However, the HAQ in 2021 is not available in public. Due to lack of the HAQ in 2021, we are unable to validate our results. We have added the limitation in line 378-379 in manuscript-revised.

Comment 2: The GDR is defined as the ratio of female-to-male QCI. A ratio close to 1 is assumed to reflect equity, but this may oversimplify interpretability. What does a GDR >1 or <1 indicate in real-world terms? Consider illustrating implications with country-level examples where GDR diverges notably from parity.

Response: We appreciate the reviewer’s insightful observation.

As defined in the study, the Gender Disparity Ratio (GDR) represents the ratio of female-to-male QCI. A value of 1 indicates parity between sexes, whereas values above 1 suggest relatively higher CKD QCI in women, and values below 1 indicate relatively higher CKD QCI in men. We have added a content your mentioned in line 172-175 and 247-286 in manuscript-revised.

Comment 3: The analysis presents decreasing YLDs alongside increasing QCI and improved GDR. Please clarify how much of the improvement in QCI reflects actual system performance versus changes in stroke incidence/severity. The limitations of indirect attribution need emphasis.

Response: We appreciate the reviewer’s insightful observation and valuable advice.

As shown in line 113-171 for QCI and line 171-175 in manuscript-revised, QCI and GDR reflects a comprehensive indicator to represent the difference in stroke care, comparing with prevalence, incidence, and mortality. Therefore, QCI and GDR reflects the whole comprehensive indicator, not a reflects the changes of prevalence, incidence, and mortality separately. In addition, the changes in stroke incidence/severity have been presented in line 181-195 in manuscript-revised.

The limitation has been added in line 377-378 in manuscript-revised.

Comment 4: While the results are rich in numerical detail, the manuscript would benefit from a concise interpretive table outlining how to read QCI and GDR values, including thresholds for high/low quality or parity gaps. This would enhance accessibility for both technical and policy audiences.

Response: We appreciate the reviewer’s insightful observation.

We have added a Table 1 in manuscript-revised.

Comment 5: Sentences such as “The situation intensifies in most regions and countries with the increase in age” (line 267) or “There was a positive association between QCI and SDI” (Abstract) are vague or grammatically unclear. Please clarify.

Response: We appreciate the reviewer’s precious suggestions.

We have revised the content in manuscript-revised.

Comment 6: While age-specific results are reported, the manuscript stops short of interpreting why older age groups exhibit different QCI or GDR patterns. Are care gaps among the elderly driven by biological vulnerability, health system neglect, or cultural inequities? A deeper exploration could enhance interpretive value.

Response: We appreciate the reviewer’s constructive advice.

We have performed the analysis of QCI and GDR focusing on older age groups. The manuscript is submitting to another journal and reviewing.

Comment 7: Phrases like “measures should be taken to improve disparities” (line 297) are too general. Consider offering more grounded recommendations, such as improving stroke recognition in women, investing in regional stroke units, or enhancing guideline adherence monitoring in low-SDI countries.

Response: Many thanks for valuable suggestions.

We have added the content you suggested in line 384-387 in manuscript-revised in manuscript-revised.

Comment 8: Given the emphasis on disparities, integrating a brief discussion of how health systems might inadvertently reinforce or mitigate inequities, especially in gender and aging populations, could enrich the policy relevance of the manuscript.

Response: We appreciate the reviewer’s insightful observation.

We have added the content in line 360-370 in manuscript-revised.

---

## [Editor Report · Decision Letter 1]

9 Feb 2026

Dear Dr. an,

Thank you for submitting your manuscript to PLOS ONE. After careful consideration, we feel that it has merit but does not fully meet PLOS ONE’s publication criteria as it currently stands. Therefore, we invite you to submit a revised version of the manuscript that addresses the points raised during the review process.

We look forward to receiving your revised manuscript.

Kind regards,

Wen-Jun Tu

Academic Editor

PLOS One

Journal Requirements:

Additional Editor Comments:

The core metric of the study, QCI, is derived from principal component analysis (PCA) of four secondary indices (MIR, DPR, PIR, YLR) and primary GBD metrics. However:The study explicitly states that validation using the Healthcare Access and Quality (HAQ) Index—an established benchmark for healthcare quality—was not possible due to the unavailability of 2021 HAQ data. Without external validation, the QCI’s ability to accurately reflect "quality of care" (as defined by the WHO) remains unsubstantiated. The authors rely on internal PCA variance explanation but provide no evidence that QCI correlates with real-world indicators of care quality (e.g., adherence to stroke treatment guidelines, hospital readmission rates, or patient-reported outcomes).

While the study correlates QCI/GDR with the Socio-Demographic Index (SDI), it does not adjust for granular factors within SDI tiers (e.g., healthcare funding, workforce density, or health insurance coverage). For example, low-SDI countries may have similar SDI scores but vastly different healthcare infrastructures, which the study fails to disentangle.

GBD showed that The annual number of strokes and deaths due to stroke increased substantially from 1990 to 2021. Data on the burden of stroke in your county should be increased so that readers can understand the need for stroke-related studies. Additional citations could be added: Estimated Burden of Stroke in China in 2020. JAMA Netw Open. 2023;6(3): e231455. doi:10.1001/jamanetworkopen.2023.1455’

---

## [Author Response · Author response to Decision Letter 2]

14 Feb 2026

Response to the Editor’s Comments

Dear Editor,

We really appreciate all of your constructive notes. All of them have helped us improve the quality of this manuscript significantly. Please find our revised manuscript in the attachments where we have addressed your notes. In the following sections, we provide point-by-point responses to your concerns.

Best regards,

Jingjing An,

MD MHA

Additional Editor Comments:

The core metric of the study, QCI, is derived from principal component analysis (PCA) of four secondary indices (MIR, DPR, PIR, YLR) and primary GBD metrics. However:The study explicitly states that validation using the Healthcare Access and Quality (HAQ) Index—an established benchmark for healthcare quality—was not possible due to the unavailability of 2021 HAQ data. Without external validation, the QCI’s ability to accurately reflect "quality of care" (as defined by the WHO) remains unsubstantiated. The authors rely on internal PCA variance explanation but provide no evidence that QCI correlates with real-world indicators of care quality (e.g., adherence to stroke treatment guidelines, hospital readmission rates, or patient-reported outcomes).

Response: Many thanks for your professional advice.

We agree that external validation against established healthcare quality metrics such as the Healthcare Access and Quality (HAQ) Index would further strengthen the interpretability of QCI. However, as noted in the manuscript, the HAQ Index data were not available for 2021, which prevented direct validation within the same study period.

Several points support the validity of QCI as a proxy for quality of care:

First, the QCI is constructed from four well-established epidemiological ratios, all of which have been widely used in previous studies as indirect indicators of healthcare effectiveness, survival, disease management, and disability burden [1,2,3,4]. These metrics reflect different dimensions of care quality, including survival outcomes, disability prevention, and long-term disease management.

In addition, it is important to emphasize that the QCI primarily reflects health outcomes and does not directly measure structural or process-related dimensions of healthcare delivery, such as adherence to stroke treatment guidelines, hospital readmission rates, or patient-reported outcomes. Moreover, the index may be influenced by factors beyond the healthcare system itself, including underlying population health status, disease ecology, and the completeness and accuracy of disease surveillance systems. Due to the limited availability of comprehensive, longitudinal data covering multiple decades and diverse geographic settings, empirical validation of the QCI against direct and independent indicators of healthcare quality remains constrained. Therefore, future research should prioritize the validation and refinement of the QCI as more granular and reliable data on healthcare delivery become available, which will further enhance its robustness and utility as a proxy measure of healthcare quality.

We have mentioned in line 400-416 in manuscript-revised.

References

1. Xu C, Zhu L, Lai J, Luo Z, Ying J, Hu S, Song P, Yang J. Global and regional quality of care index in major depressive disorder: the global burden of disease study 2021. Int J Equity Health. 2026 Feb 7. doi: 10.1186/s12939-026-02775-5. Epub ahead of print. PMID: 41654951.

2. Xia Z, Wang J, Wen X. Global assessment of pain care quality for musculoskeletal disorders: Insights from the quality of care index (QCI) from 1990 to 2021. PLoS One. 2025 Oct 31;20(10):e0335235. doi: 10.1371/journal.pone.0335235. PMID: 41171723;

3. Lin S, Liu Q, Yin L, Liu W, Zhu X, Shen Y, Li Z, Feng B. Global, regional, and national trends and inequality analysis of the total cancers and 29 cancer types from 1990 to 2021. BMC Public Health. 2025 Oct 15;25(1):3494. doi: 10.1186/s12889-025-24671-3. PMID: 41094448;

4. He R, Zhu W, Hui S, Yu M, Li H, Li Y, Huang P, Yu R. Inequalities in disease burden and care quality of neglected tropical diseases and malaria, 1990-2021: Findings from the 2021 Global Burden of Disease Study. PLoS One. 2025 Aug 7;20(8):e0329475. doi: 10.1371/journal.pone.0329475. PMID: 40773432; P

While the study correlates QCI/GDR with the Socio-Demographic Index (SDI), it does not adjust for granular factors within SDI tiers (e.g., healthcare funding, workforce density, or health insurance coverage). For example, low-SDI countries may have similar SDI scores but vastly different healthcare infrastructures, which the study fails to disentangle.

Response: Many thanks for your professional advice.

Thank you for this insightful comment. We fully agree that countries within the same SDI category may differ substantially in specific healthcare system characteristics, such as healthcare expenditure, workforce density, insurance coverage, and infrastructure capacity, which are not fully captured by the composite SDI measure.

However, SDI is a well-established composite indicator developed within the Global Burden of Disease (GBD) framework, incorporating income per capita, educational attainment, and fertility rate, and has been widely used as a proxy for overall socioeconomic development and health system capacity in global health research [1,2,3]. The use of SDI in this study was intended to examine broad sociodemographic gradients in quality of care rather than to isolate the independent effects of specific healthcare system components.

Importantly, the primary aim of this study was to characterize global, regional, and national patterns and disparities in stroke care quality, rather than to establish causal relationships between healthcare system inputs and quality outcomes. Given the global scope of the analysis and the inclusion of 204 countries and territories over multiple decades, consistent and comparable data on granular healthcare system variables, such as healthcare expenditure, workforce density, and insurance coverage, are not uniformly available across all locations and time periods.

We acknowledge that residual heterogeneity in healthcare system characteristics may exist within SDI strata. This limitation has now been explicitly acknowledged in line 416-430 in manuscript-revised. Future studies incorporating more detailed healthcare system indicators will be essential to further disentangle the relative contributions of specific healthcare resources and policies to variations in care quality.

References

1. GBD 2019 Diseases and Injuries Collaborators. Global burden of 369 diseases and injuries in 204 countries and territories, 1990-2019: a systematic analysis for the Global Burden of Disease Study 2019. Lancet. 2020 Oct 17;396(10258):1204-1222. doi: 10.1016/S0140-6736(20)30925-9. Erratum in: Lancet. 2020 Nov 14;396(10262):1562. doi: 10.1016/S0140-6736(20)32226-1. PMID: 33069326;

2. GBD 2021 Asthma and Allergic Diseases Collaborators. Global, regional, and national burden of asthma and atopic dermatitis, 1990-2021, and projections to 2050: a systematic analysis of the Global Burden of Disease Study 2021. Lancet Respir Med. 2025 May;13(5):425-446. doi: 10.1016/S2213-2600(25)00003-7. Epub 2025 Mar 24. PMID: 40147466.

3. GBD 2021 Adult BMI Collaborators. Global, regional, and national prevalence of adult overweight and obesity, 1990-2021, with forecasts to 2050: a forecasting study for the Global Burden of Disease Study 2021. Lancet. 2025 Mar 8;405(10481):813-838. doi: 10.1016/S0140-6736(25)00355-1.

GBD showed that The annual number of strokes and deaths due to stroke increased substantially from 1990 to 2021. Data on the burden of stroke in your county should be increased so that readers can understand the need for stroke-related studies. Additional citations could be added: Estimated Burden of Stroke in China in 2020. JAMA Netw Open. 2023;6(3): e231455.’

Response: Many thanks for your professional advice.

We have added the content in line 295-304 and 323-327 in manuscript-revised.

---

## [Editor Report · Decision Letter 2]

10 Mar 2026

Dear Dr. an,

Thank you for submitting your manuscript to PLOS ONE. After careful consideration, we feel that it has merit but does not fully meet PLOS ONE’s publication criteria as it currently stands. Therefore, we invite you to submit a revised version of the manuscript that addresses the points raised during the review process.

We look forward to receiving your revised manuscript.

Kind regards,

Wen-Jun Tu

Academic Editor

PLOS One

**Journal Requirements:**

**Additional Editor Comments:**

The references are extremely disorganized.

---

## [Author Response · Author response to Decision Letter 3]

11 Mar 2026

Response to the Editor’s Comments

Dear Editor,

We really appreciate all of your constructive notes. All of them have helped us improve the quality of this manuscript significantly. Please find our revised manuscript in the attachments where we have addressed your notes. In the following sections, we provide point-by-point responses to your concerns.

Best regards,

Jingjing An,

MD MHA

Additional Editor Comments:

The core metric of the study, QCI, is derived from principal component analysis (PCA) of four secondary indices (MIR, DPR, PIR, YLR) and primary GBD metrics. However:The study explicitly states that validation using the Healthcare Access and Quality (HAQ) Index—an established benchmark for healthcare quality—was not possible due to the unavailability of 2021 HAQ data. Without external validation, the QCI’s ability to accurately reflect "quality of care" (as defined by the WHO) remains unsubstantiated. The authors rely on internal PCA variance explanation but provide no evidence that QCI correlates with real-world indicators of care quality (e.g., adherence to stroke treatment guidelines, hospital readmission rates, or patient-reported outcomes).

Response: Many thanks for your professional advice.

We agree that external validation against established healthcare quality metrics such as the Healthcare Access and Quality (HAQ) Index would further strengthen the interpretability of QCI. However, as noted in the manuscript, the HAQ Index data were not available for 2021, which prevented direct validation within the same study period.

Several points support the validity of QCI as a proxy for quality of care:

First, the QCI is constructed from four well-established epidemiological ratios, all of which have been widely used in previous studies as indirect indicators of healthcare effectiveness, survival, disease management, and disability burden [1,2,3,4]. These metrics reflect different dimensions of care quality, including survival outcomes, disability prevention, and long-term disease management.

In addition, it is important to emphasize that the QCI primarily reflects health outcomes and does not directly measure structural or process-related dimensions of healthcare delivery, such as adherence to stroke treatment guidelines, hospital readmission rates, or patient-reported outcomes. Moreover, the index may be influenced by factors beyond the healthcare system itself, including underlying population health status, disease ecology, and the completeness and accuracy of disease surveillance systems. Due to the limited availability of comprehensive, longitudinal data covering multiple decades and diverse geographic settings, empirical validation of the QCI against direct and independent indicators of healthcare quality remains constrained. Therefore, future research should prioritize the validation and refinement of the QCI as more granular and reliable data on healthcare delivery become available, which will further enhance its robustness and utility as a proxy measure of healthcare quality.

We have mentioned in line 400-416 in manuscript-revised.

References

1. Xu C, Zhu L, Lai J, Luo Z, Ying J, Hu S, Song P, Yang J. Global and regional quality of care index in major depressive disorder: the global burden of disease study 2021. Int J Equity Health. 2026 Feb 7. doi: 10.1186/s12939-026-02775-5. Epub ahead of print. PMID: 41654951.

2. Xia Z, Wang J, Wen X. Global assessment of pain care quality for musculoskeletal disorders: Insights from the quality of care index (QCI) from 1990 to 2021. PLoS One. 2025 Oct 31;20(10):e0335235. doi: 10.1371/journal.pone.0335235. PMID: 41171723;

3. Lin S, Liu Q, Yin L, Liu W, Zhu X, Shen Y, Li Z, Feng B. Global, regional, and national trends and inequality analysis of the total cancers and 29 cancer types from 1990 to 2021. BMC Public Health. 2025 Oct 15;25(1):3494. doi: 10.1186/s12889-025-24671-3. PMID: 41094448;

4. He R, Zhu W, Hui S, Yu M, Li H, Li Y, Huang P, Yu R. Inequalities in disease burden and care quality of neglected tropical diseases and malaria, 1990-2021: Findings from the 2021 Global Burden of Disease Study. PLoS One. 2025 Aug 7;20(8):e0329475. doi: 10.1371/journal.pone.0329475. PMID: 40773432; P

While the study correlates QCI/GDR with the Socio-Demographic Index (SDI), it does not adjust for granular factors within SDI tiers (e.g., healthcare funding, workforce density, or health insurance coverage). For example, low-SDI countries may have similar SDI scores but vastly different healthcare infrastructures, which the study fails to disentangle.

Response: Many thanks for your professional advice.

Thank you for this insightful comment. We fully agree that countries within the same SDI category may differ substantially in specific healthcare system characteristics, such as healthcare expenditure, workforce density, insurance coverage, and infrastructure capacity, which are not fully captured by the composite SDI measure.

However, SDI is a well-established composite indicator developed within the Global Burden of Disease (GBD) framework, incorporating income per capita, educational attainment, and fertility rate, and has been widely used as a proxy for overall socioeconomic development and health system capacity in global health research [1,2,3]. The use of SDI in this study was intended to examine broad sociodemographic gradients in quality of care rather than to isolate the independent effects of specific healthcare system components.

Importantly, the primary aim of this study was to characterize global, regional, and national patterns and disparities in stroke care quality, rather than to establish causal relationships between healthcare system inputs and quality outcomes. Given the global scope of the analysis and the inclusion of 204 countries and territories over multiple decades, consistent and comparable data on granular healthcare system variables, such as healthcare expenditure, workforce density, and insurance coverage, are not uniformly available across all locations and time periods.

We acknowledge that residual heterogeneity in healthcare system characteristics may exist within SDI strata. This limitation has now been explicitly acknowledged in line 416-430 in manuscript-revised. Future studies incorporating more detailed healthcare system indicators will be essential to further disentangle the relative contributions of specific healthcare resources and policies to variations in care quality.

References

1. GBD 2019 Diseases and Injuries Collaborators. Global burden of 369 diseases and injuries in 204 countries and territories, 1990-2019: a systematic analysis for the Global Burden of Disease Study 2019. Lancet. 2020 Oct 17;396(10258):1204-1222. doi: 10.1016/S0140-6736(20)30925-9. Erratum in: Lancet. 2020 Nov 14;396(10262):1562. doi: 10.1016/S0140-6736(20)32226-1. PMID: 33069326;

2. GBD 2021 Asthma and Allergic Diseases Collaborators. Global, regional, and national burden of asthma and atopic dermatitis, 1990-2021, and projections to 2050: a systematic analysis of the Global Burden of Disease Study 2021. Lancet Respir Med. 2025 May;13(5):425-446. doi: 10.1016/S2213-2600(25)00003-7. Epub 2025 Mar 24. PMID: 40147466.

3. GBD 2021 Adult BMI Collaborators. Global, regional, and national prevalence of adult overweight and obesity, 1990-2021, with forecasts to 2050: a forecasting study for the Global Burden of Disease Study 2021. Lancet. 2025 Mar 8;405(10481):813-838. doi: 10.1016/S0140-6736(25)00355-1.

GBD showed that The annual number of strokes and deaths due to stroke increased substantially from 1990 to 2021. Data on the burden of stroke in your county should be increased so that readers can understand the need for stroke-related studies. Additional citations could be added: Estimated Burden of Stroke in China in 2020. JAMA Netw Open. 2023;6(3): e231455.’

Response: Many thanks for your professional advice.

We have added the content in line 295-304 and 323-327 in manuscript-revised.

---

## [Editor Report · Decision Letter 3]

13 Mar 2026

Quality of Care Index and Gender disparity ratio for Stroke and its subtypes from the Global Burden of Disease Study 2021

PONE-D-25-27376R3

Dear Dr. an,

We’re pleased to inform you that your manuscript has been judged scientifically suitable for publication and will be formally accepted for publication once it meets all outstanding technical requirements.

Kind regards,

Wen-Jun Tu

Academic Editor

PLOS One
---

## [Editor Report · Acceptance letter]

PONE-D-25-27376R3

PLOS One

Dear Dr. an,

I'm pleased to inform you that your manuscript has been deemed suitable for publication in PLOS One. Congratulations! Your manuscript is now being handed over to our production team.

Kind regards,

on behalf of

Dr. Wen-Jun Tu

Academic Editor

PLOS One